# Investigation of the Influence of Polyamines on Mature Embryo Culture and DNA Methylation of Wheat (*Triticum aestivum* L.) Using the Machine Learning Algorithm Method

**DOI:** 10.3390/plants12183261

**Published:** 2023-09-13

**Authors:** Barış Eren, Aras Türkoğlu, Kamil Haliloğlu, Fatih Demirel, Kamila Nowosad, Güller Özkan, Gniewko Niedbała, Alireza Pour-Aboughadareh, Henryk Bujak, Jan Bocianowski

**Affiliations:** 1Department of Agricultural Biotechnology, Faculty of Agriculture, Igdır University, Igdir 76000, Türkiye; baris.eren@igdir.edu.tr (B.E.); fatih.demirel@igdir.edu.tr (F.D.); 2Department of Field Crops, Faculty of Agriculture, Necmettin Erbakan University, Konya 42310, Türkiye; 3Department of Field Crops, Faculty of Agriculture, Ataturk University, Erzurum 25240, Türkiye; kamilh@atauni.edu.tr; 4Department of Genetics, Plant Breeding and Seed Production, Wrocław University of Environmental and Life Sciences, Grunwaldzki 24A, 53-363 Wrocław, Poland; h.bujak@coboru.gov.pl; 5Department of Biology, Faculty of Science, Ankara University, Ankara 06100, Türkiye; gullerozzkan@gmail.com; 6Department of Biosystems Engineering, Faculty of Environmental and Mechanical Engineering, Poznan University of Life Sciences, Wojska Polskiego 50, 60-627 Poznań, Poland; gniewko.niedbala@up.poznan.pl; 7Seed and Plant Improvement Institute, Agricultural Research, Education and Extension Organization (AREEO), Karaj P.O. Box 3158854119, Iran; a.poraboghadareh@edu.ikiu.ac.ir; 8Research Centre for Cultivar Testing (COBORU), Słupia Wielka 34, 63-022 Słupia Wielka, Poland; 9Department of Mathematical and Statistical Methods, Poznan University of Life Sciences, Wojska Polskiego 28, 60-637 Poznań, Poland

**Keywords:** DNA methylation, genomic template stability, iPBS, machine learning

## Abstract

Numerous factors can impact the efficiency of callus formation and *in vitro* regeneration in wheat cultures through the introduction of exogenous polyamines (PAs). The present study aimed to investigate *in vitro* plant regeneration and DNA methylation patterns utilizing the inter-primer binding site (iPBS) retrotransposon and coupled restriction enzyme digestion–iPBS (CRED–iPBS) methods in wheat. This investigation involved the application of distinct types of PAs (Put: putrescine, Spd: spermidine, and Spm: spermine) at varying concentrations (0, 0.5, 1, and 1.5 mM). The subsequent outcomes were subjected to predictive modeling using diverse machine learning (ML) algorithms. Based on the specific polyamine type and concentration utilized, the results indicated that 1 mM Put and Spd were the most favorable PAs for supporting endosperm-associated mature embryos. Employing an epigenetic approach, Put at concentrations of 0.5 and 1.5 mM exhibited the highest levels of genomic template stability (GTS) (73.9%). Elevated Spd levels correlated with DNA hypermethylation while reduced Spm levels were linked to DNA hypomethylation. The *in vitro* and epigenetic characteristics were predicted using ML techniques such as the support vector machine (SVM), extreme gradient boosting (XGBoost), and random forest (RF) models. These models were employed to establish relationships between input variables (PAs, concentration, GTS rates, *Msp* I polymorphism, and *Hpa* II polymorphism) and output parameters (*in vitro* measurements). This comparative analysis aimed to evaluate the performance of the models and interpret the generated data. The outcomes demonstrated that the XGBoost method exhibited the highest performance scores for callus induction (CI%), regeneration efficiency (RE), and the number of plantlets (NP), with R^2^ scores explaining 38.3%, 73.8%, and 85.3% of the variances, respectively. Additionally, the RF algorithm explained 41.5% of the total variance and showcased superior efficacy in terms of embryogenic callus induction (ECI%). Furthermore, the SVM model, which provided the most robust statistics for responding embryogenic calluses (RECs%), yielded an R^2^ value of 84.1%, signifying its ability to account for a substantial portion of the total variance present in the data. In summary, this study exemplifies the application of diverse ML models to the cultivation of mature wheat embryos in the presence of various exogenous PAs and concentrations. Additionally, it explores the impact of polymorphic variations in the CRED–iPBS profile and DNA methylation on epigenetic changes, thereby contributing to a comprehensive understanding of these regulatory mechanisms.

## 1. Introduction

Polyamines (PAs), possessing elevated biological activity, constitute a class of low molecular weight aliphatic nitrogenous organic compounds harboring two or more amino groups [1]. Among these, putrescine (Put), spermine (Spm), and spermidine (Spd) stand as the most prevalent forms of polyamines. Their significance is underscored by their pivotal roles in an array of biological processes, encompassing tissue growth, cell division, cell differentiation, and programmed cell death [2]. Furthermore, polyamines exhibit notable involvement in responding to both biotic and abiotic stresses [3], which constitute just a fragment of their multifaceted physiological functions. Of specific note is their pronounced impact on somatic embryogenesis across diverse plant species [4]. Recent investigations have illuminated the pivotal role of polyamine metabolism in the context of somatic embryogenesis within wheat. Notably, the collective balance of endogenous polyamine levels and the ratios among individual polyamine species have been identified as critical factors during the initial phases of somatic embryogenesis [5,6]. To engineer a triumphant *in vitro* regeneration strategy, meticulous modulation of input parameters within plant tissue culture becomes imperative [7]. Following the induction of callus formation, the subsequent phase of regeneration establishment emerges as a pivotal stride in realizing the ultimate success of plant tissue culture endeavors. Deeper insights into the intricacies of callus induction and the subsequent regeneration process are attainable through the adept application of plant tissue culture methodologies.

Profoundly dynamic processes, referred to as epigenetic mechanisms, intricately orchestrate the regulation of gene expression. The comprehensive analysis of the epigenetic landscape within plant cell cultures assumes increasing significance, particularly as more sophisticated and potent epigenetic analysis technologies become accessible, spanning a wider array of plant species [8]. In the realm of plant biology, the spectrum of epigenetic variations encompasses diverse phenomena, such as point mutations, deletions, transposable element activations, rearrangements, and changes in ploidy [9,10], as well as alterations to the DNA structure itself [11]. Various molecular markers, including single nucleotide polymorphisms (SNPs), microsatellites (SSRs), simple sequence repeat polymorphisms (ISSRs), fragment length polymorphisms (RFLPs), and random amplified polymorphic DNA (RAPD) markers [12], alongside the inter-primer binding site (iPBS) retrotransposon [13,14], are currently harnessed for detecting polymorphic states and DNA methylation patterns [15,16,17,18]. Within the context of epigenetic modulation, DNA methylation emerges as a pivotal process, universally governing gene expression and the repression of transposable elements. This process involves the addition of a methyl group to the fifth position of cytosine. Remarkably, DNA methylation stands as an inheritable yet reversible phenomenon, as the methyl group can be enzymatically removed. A pivotal method in the precise investigation of DNA methylation in plants is the coupled restriction enzyme digestion (CRED) technique [14]. Through the CRED methodology, DNA profiling is facilitated utilizing an ensemble of molecular markers. Furthermore, the integration of linked restriction enzyme digestion and inter-primer binding site analysis, known as CRED–iPBS, stands as an indispensable approach for investigating the methylation status of plant genes. This technique discerns alterations in cytosine methylation patterns within plant genomes that arise due to environmental stressors [19]. A plethora of studies have collectively elucidated the multifaceted roles of polyamines. These include safeguarding DNA replication from oxidative damage, stimulating DNA and RNA biosynthesis [20], and more intriguingly, selectively modulating cytosine DNA methylation through intricate interactions and binding mechanisms [21].

The utilization of diverse machine learning (ML) algorithm models within the domain of plant biotechnology stands as a recent and burgeoning area of exploration aimed at predicting and optimizing variables within intricate biological systems [22,23]. ML serves as a paradigm of data science that confronts intricate challenges spanning diverse scientific disciplines. This approach surpasses conventional unidirectional analyses, facilitating a nuanced comprehension and precise interpretation of results [24]. The application of ML has thus far demonstrated successful outcomes across various domains of plant science, encompassing *in vitro* germination [22,24], regeneration studies [22,25,26,27,28], *in vitro* mutagenesis [29], and more. In the landscape of ML algorithms, diverse models rooted in artificial intelligence principles, coupled with an array of performance metrics, are harnessed to validate predicted outcomes. Among these models, the support vector machine (SVM) emerges as a widely adopted machine learning algorithm, adeptly addressing both classification and regression tasks [30]. The random forest (RF) algorithm, an ensemble learning methodology, orchestrates the amalgamation of numerous decision trees [31]. Likewise, XGBoost, renowned for extreme gradient boosting, epitomizes a gradient-boosting technique applicable to the realms of both classification and regression tasks [32]. The current era is witnessing a confluence of advanced computational techniques with plant biotechnology, offering novel avenues for insights into intricate biological phenomena and the prospect of refining interventions within this complex domain.

Contemporary investigations across a spectrum of plant species necessitate the incorporation of current insights into *in vitro* plant tissue culture technologies and the intricate tapestry of epigenetic variations. Grasping the intricate ramifications of diverse hormones, such as auxins and cytokinins, along with polyamines, within the context of plant tissue culture and DNA methylation, particularly in the paradigm of wheat tissue culture, is a formidable undertaking. The formulation of informed decisions based on scientific findings in this context constitutes a challenging endeavor. This challenge, however, can be mitigated through the strategic implementation of diverse models and algorithms, thereby elevating the precision and accuracy of predictive assessments. The application of mathematical frameworks and artificial intelligence-based models under *in vitro* conditions, with a focus on comprehending the intricate dynamic of callus induction, regeneration efficiency, and the DNA methylation process, remains conspicuously constrained within the scope of wheat. The objectives of this study were: (1) to investigate the effects of different PAs on *in vitro* regeneration; (2) to detect changes in DNA methylation using the CRED method applied to iPBS markers; and (3) to statistically analyze and then validate the data through supervised machine learning modeling of *in vitro* regeneration and wheat DNA methylation in different PAs.

## 2. Results

### 2.1. In Vitro Parameters

Plant tissue culture research requires the efficient development of *in vitro* conditions. Polyamines (PAs) are a class of naturally occurring chemical compounds that play an important role in cell growth and development and in the stress response to a wide range of environmental stresses. The results of the analysis of variance showed that the main effect of the type and concentration of PAs was significant for all parameters evaluated (Table 1). Although callus induction from endosperm-assisted maturation began almost simultaneously for all types of PAs, the rate of callus induction was most significant for Put concentrations of 0.5 and 1 mM. The highest callus induction was observed in Murashige and Skoog (MS) medium supplemented with Put 1 mM (97.50%), while the lowest callus induction occurred in medium containing Spm 1 mM (83.75%). Callus induction ranged from 87.50% to 93.75% depending on different types and concentrations of polyamine. Moreover, the callus induction rate decreased with increasing polyamine concentrations. The most suitable concentrations for callus induction were 0.5 mM and 1 mM for Put and Spd (Table 1).

The induction of embryogenic callus under the influence of different polyamine types and concentrations is shown in Table 1. According to the data obtained, the highest induction embryogenic callus (100.00%) in MS medium was obtained with Spd 1.5 mM treatment. As the concentrations of Put and Spd media increased, the rate of embryogenic callus also increased. However, there was a decrease in the embryogenic callus rate with Spm compared to the control group.

A plant’s ability to develop embryogenic callus is linked to its ability to regenerate. However, not all embryos and embryogenic calluses will develop into fully functional, regenerating plants. Therefore, we identified embryogenic calluses that gave rise to roots and shoots as responsive embryogenic calluses (RECs) and calculated RE. Spd type and concentration were more effective than Put and Spm for RECs. The lowest REC was 35% at an Spm concentration of 1.5 mM (Table 1).

The number of regenerated plants per explant (RE) under different types and concentrations of polyamine ranged from 0.98 to 1.19. The highest RE was observed with a Put treatment of 1.5 mM (1.29). The lowest RE value was obtained with Spm 1.5 mM treatment (0.37). Increasing polyamine concentration led to an increase in Put treatment, while Spm treatment led to a decrease. In the interaction between polyamine type and concentration, Put and Spd were the most effective polyamines in wheat endosperm-assisted plant regeneration through mature culture (Table 1).

The average count of regenerated plants manifested discernible responsiveness to the interplay of polyamine types and concentrations. The calculated average tally of regenerated plants per explant exhibited a spectrum spanning from 2.0 to 12.25, contingent upon the distinct polyamine types and their corresponding concentrations. Notably, among the diverse polyamines investigated, the culture medium supplemented with 1 mM putrescine yielded the most pronounced effect, yielding the highest mean count of regenerated plants per explant, at 12.25. The observed variance in the mean count of regenerated plants was primarily governed by the diverse concentrations of polyamines. Specifically, the regimen featuring 1.5 mM spermine registered the lowest mean regeneration efficiency, with a count of 2.0 plants per explant. Upon meticulous scrutiny of the data, it becomes evident that a reduction in the concentrations of polyamines engenders an augmentation in the count of regenerated plants, as delineated in Table 1.

### 2.2. iPBS Analysis

A multitude of fundamental cellular processes, including, but not limited to, DNA replication, transcription, translation, cell proliferation, modulation of enzyme activity, maintenance of cellular cation balance, and preservation of membrane integrity, fall under the regulatory purview of polyamines (PAs). Within the realm of this investigation, the intricate genetic and epigenetic ramifications ensuing from the application of diverse polyamine types and concentrations within wheat plants were systematically elucidated through the employment of iPBS and CRED–iPBS methodologies. The selection of iPBS primers was executed with precision and guided by their capability to engender distinctive and readily quantifiable band patterns (Figure 1).

The amassed data unveiled discernible alterations within wheat plants when subjected to distinct polyamine types and concentrations. The investigation encompassed an evaluation of the total count of polymorphic bands, which stood at 69 in the control iPBS group (as expounded upon in Table 2). The polymorphic prevalence within varying polyamine types exhibited a range spanning from 26.10% to 29.00% for Put (at concentrations of 0.5, 1.0, and 1.5 mM), 30.40% to 37.70% for Spd (across concentrations of 0.5, 1.0, and 1.5 mM), and 27.5% to 43.5% for Spm (across concentrations of 0.5, 1.0, and 1.5 mM), as graphically delineated in Figure 2A.

The polymorphism rates exhibited variability contingent upon the distinct polyamine types and concentrations under examination. Notably, the most conspicuous polymorphism rate was recorded at 43.50% for Spm at a concentration of 1.5 mM, closely followed by a rate of 37.70% observed at an Spd concentration of 0.5 mM. Furthermore, a notable polymorphism rate of 34.80% manifested at an Spm concentration of 0.5 mM (Figure 2B).

In accordance with distinct variations in polyamine types and concentrations, the most prominent genomic template stability (GTS) percentage was ascertained to be 73.90% within the Put group (at concentrations of 0.5 and 1.5 mM). Conversely, the lowest GTS percentage of 62.30% was observed within the Spd group at a concentration of 0.5 mM. Corroborating the gleaned dataset, a discernible pattern emerged wherein an escalation in concentration correlated with a diminishment in GTS percentages and an elevation in polymorphism rates. This trend is exemplified in the case of 1.5 mM Spm, wherein a polymorphism rate of 43.5% was registered, contrasting the lowest GTS percentage of 56.5% (Figure 2C).

Within the purview of the CRED–iPBS investigation, a comprehensive set of ten distinct primers was judiciously employed to elucidate diverse cytosine methylation patterns embedded within the genomic DNA (gDNA). The outcomes derived from the CRED–iPBS research are concisely consolidated as the mean percentage of cytosine methylation polymorphism associated with each concentration level (Table 3). These averages encapsulate the polymorphism percentages of cytosine methylation, unraveling the nuanced dynamics across diverse polyamine types and concentrations, thereby furnishing the cornerstone of CRED–iPBS insights. The results of the CRED–iPBS assay are indicated as the percentage of polymorphisms in CRED–iPBS assays that were digested by *Msp* I and *Hpa* II (Figure 3). Notably, discerning from the accumulated data, the maximal polymorphism rate attributed to *Msp* I digestion was attained within the ambit of Spd treatment at 1.5 mM, amounting to 52.50%. In contrast, the nadir of polymorphism was documented within Spm treatment at a concentration of 0.5 mM, registering at 29.51%. Within the diverse polyamine types and concentrations, a total of 94 novel bands were distinctly identified in association with *Msp* I relative to the control group. Additionally, 126 pre-existing bands were observed to abate under the influence of *Msp* I (Table 3 and Figure 4).

The highest polymorphism rate for *Hpa* II was 42.40% in the Spm 0.5 Mm treatment, while the lowest value was 23.7% in the Put 0.5 Mm treatment. For *Hpa* II, 86 new bands appeared compared to the control group. In addition, 116 old bands disappeared for *Msp* I. According to the data obtained, it was determined that the rate of polymorphism due to the *Msp* I enzyme was higher than *Hpa* II (Table 3 and Figure 4).

### 2.3. Machine Learning (ML) Analysis

The process of machine learning algorithms entails the utilization of statistical and computational methodologies to extract patterns from data, construct predictive models, and subsequently employ these models for making informed predictions or decisions. In the context of this study, the support vector machine (SVM), the random forest (RF) algorithm, and the extreme gradient boosting (XGBoost) approach were deployed to anticipate the intricate relationships between input variables and output parameters. This exercise encompassed a comprehensive evaluation and comparative analysis of model performances guided by the need to dissect the intricate dataset emerging from both tissue culture and molecular analyses. For the purpose of model training and validation, the dataset was systematically partitioned into two subsets: a training dataset, constituting 70.00% of the overall dataset, and a separate testing dataset accounting for the remaining 30.00%. The model’s efficacy was assessed through meticulous testing against the reserved test dataset while the training dataset facilitated the iterative learning process employed by the model. This approach enables the extrapolation of insights from smaller datasets to predict outcomes in larger datasets, thereby enhancing the generalizability of the model. The input variables in this study encompassed three discrete polyamines under experimental conditions (Put, Spd, and Spm) alongside the corresponding range of polyamine concentrations (0, 0.5, 1, and 1.5 mM). Additionally, the input dataset integrated the outcomes of genomic template stability (GTS) rates, *Msp* I polymorphism, and *Hpa* II polymorphism within the context of wheat, each influenced by the experimental configurations. Consequently, the predictions concerning the observed variables (CI, ECI, REC, RE, PN) were derived as a cumulative outcome of the intricate interplay between exogenously administered polyamines and the resultant genomic DNA variations. The tabulated outcomes of the machine learning models employed in this investigation are meticulously presented in Table 4.

The evaluation of algorithm performance is predicated on metrics encompassing mean squared error (MSE), mean absolute percentage error (MAPE), and mean absolute deviation (MAD). Reduced values of these metrics signify a closer alignment between the model’s prognostications and the actual observed values. Upon scrutinizing the test performance outcomes of MSE, MAPE, and MAD, distinct trends surfaced: XGBoost < RF < SVM for CI, RE, and PN. For the ECI variable, the sequence of RF < XGBoost < SVM emerged based on the MSE, MAPE, and MAD results, whereas for REC, the order was SVM < RF < XGBoost. Delving into the R^2^ outcomes furnished by the XGBoost algorithm, renowned for yielding optimal performance outcomes concerning CI, RE, and PN, the percentages of variance elucidated were 38.30%, 73.80%, and 85.30%, respectively. Meanwhile, the R^2^ result derived from the RF algorithm accentuated its explanation of 41.50% of the overall variation, showcasing superior performance, especially with regards to ECI. Further, the R^2^ outcome from the SVM algorithm, which attained the highest performance metrics for REC, distinctly encapsulated 84.10% of the comprehensive variance evident in the dataset.

## 3. Discussion

This research endeavors to elucidate the influence of diverse polyamine species and varying concentrations thereof on the initiation of callus formation and the ensuing regenerative processes in plants. The response rate of responded embryogenic calluses (RECs) and the propensities of mature embryos to undergo callus induction exhibited discernible alterations in response to polyamine concentrations. Notably, Spd treatment at 1.5 mM engendered a 100.00% rate of embryogenic callus induction, although a subsequent concentration of 1.5 mM led to a decrement of 35% in REC. It was evident that the frequency of embryogenic callus formation exhibited a direct correlation with increasing levels of Put and Spd, while Spm administration was characterized by a lowered rate compared to the control group. The regeneration efficiency (RE) values for the diverse polyamines and their respective concentrations ranged between 0.98 and 1.19. Significantly, the most notable RE was achieved through Put treatment at 1.5 mM, attaining a value of 1.29, while a RE value of 0.37 was obtained in the context of Spm treatment at 1.5 mM. Notably, Put treatment exhibited an augmenting trend with increasing polyamine concentrations, whereas the converse was evident in the case of Spm treatment. The interaction between polyamine type and concentration underscored that the most favorable conditions for callus induction were associated with 0.5 mM of Put and 1 mM of spermidine. Elevated regenerative potential was observed in tandem with augmented levels of Put and spermidine within somatic embryos and shoots [33]. The discerned roles of Spm and Spd come to the fore when developmental progression is impeded, while Put assumes significance in the early embryogenic stages characterized by heightened cellular division rates [34]. Notably, Put emerges as the predominant polyamine subtype, succeeded by Spd and Spm [35]. These findings are fortified by antecedent research conducted by Martinez et al. [36], Dewi et al. [37], Tang et al. [38], Li et al. [39], Purwoko et al. [40], Paul et al. [41], Aydin et al. [5], Rakesh et al. [42], Xiong et al. [43], and Liu et al. [44]. The supplementation of the regeneration medium with polyamines, specifically spermidine and Put, during *in vitro* induction demonstrated improved gynogenetic embryo and haploid plantlet development [45]. Further evidence in studies indicates that the potency of spermine and spermidine in enhancing somatic embryogenesis and plant regeneration is notably observed in lower dosages in comparison to putrescine.

DNA methylation, histone methylation/demethylation modifications, and the expression of short RNAs all bear pivotal significance in the context of plant tissue culture [46,47,48,49]. Furthermore, a discernible void exists within *in vitro* studies pertaining to the intricate interplay of polyamines with DNA damage in *Triticum aestivum* L. It is noteworthy that the impact of polyamines on DNA methylation in plants remains relatively unexplored, as per the existing corpus of literature. As such, the primary thrust of this study dissected the ramifications of polyamines on DNA damage, polymorphism, and genomic stability within the milieu of wheat plants under *in vitro* conditions. The study by Temel et al. [50] propounds that DNA methylation is optimally elucidated within tissue culture research through the prism of the CRED methodology. The amalgamation of iPBS and CRED methodologies [14,15,16,17,18] constitutes a boon for tissue culture investigations, empowering the delineation of both genetic and epigenetic transformations. In the current research, the synergy of iPBS and CRED was harnessed to unravel the polyamine-induced perturbations in the genetic and epigenetic landscapes of wheat. Within this paradigm, polymorphism is encapsulated by the emergence of novel bands or the vanishing of canonical bands, with point mutations, insertions, and deletions at loci attributed to the employed markers representing potential causal factors. Cumulatively, the iPBS control group exhibited a total of 69 polymorphic bands. The dynamic spectrum of polymorphic bands spanned from 26.10% to 26.60% for Put (0.5 to 1.0 to 1.5 mM), 30.30% to 37.70% for Spd (0.5 to 1.0 to 1.5 mM), and 27.50% to 43.50% for Spm (0.5 to 1.0 to 1.5 mM). Diverging from the discerned augmentation in GTS values with heightened concentrations in the cases of Put and Spd, Spm exhibited an inverse correlation. While the GTS value demonstrated an attenuation concomitant with escalated concentration, the highest polymorphism rate (43.50%) transpired within the Spm treatment. The discerned outcomes collectively hint at the preferential attributes of putrescine, as it emerges as a safeguard against the deleterious impacts of *in vitro* medium-induced stress on wheat DNA, with demethylation exerting a favorable influence on stress tolerance. DNA methylation emerges as a pivotal enzymatic modification, engendering chemical transformations within the DNA molecule. This, in turn, substantiates its cardinal role in the regulation of gene expression and the preservation of genomic integrity [51]. Despite the escalated polymorphism rate, this study reveals the ability of diverse polyamines and their concentrations to impede the retrotransposon migration process. In the context of potential adversities impinging on plants, such as heightened polyamine levels [52], plants might instigate molecular defense mechanisms. Elevated Spm content, as suggested by the diminished GTS ratio, bears the potential imprint of genotoxic consequences. Intriguingly, polyamines not only shield DNA from enzymatic degradation and X-ray irradiation but also confer stability to RNA, thereby inhibiting ribosomal dispersal [21]. Moreover, polyamines have been reported to fortify replicating DNA against oxidative harm and to expedite the biosynthesis of DNA and RNA, as evidenced by certain studies [20]. The protective attributes of putrescine might conceivably be attributed to its positively charged nature. A recent exploration of methylation in maize plants disclosed a decline in GTS rates [53]. The iPBS methodology emerges as a sentinel for DNA damage, aptly detecting shifts in the iPBS profile through the lens of a GTS rate downturn vis-à-vis a control group [54]. The outcomes unravel a dynamic: DNA hypermethylation is linked to elevated Spd levels and DNA hypomethylation is associated with diminished Spm levels. By unequivocally demonstrating hypomethylation arising from subdued polyamine concentrations, this study contributes to a robust understanding of the dichotomy. Evidently, hypermethylation herald’s gene silencing, while hypomethylation heralds heightened transcriptional activity [55]. The holistic spectrum of epigenetic alterations underscores how cells undergo differentiation in response to environmental cues [56]. Cellular differentiation manifests through distinctive expressions of epigenetic hallmarks such as DNA methylation and histone modifications [57,58]. Reflecting on the prevailing findings, the discerned protective attributes of putrescine could conceivably be attributed to its prowess in radical scavenging, potent antioxidant activities, and fundamental net charge.

This investigation encompassed the pursuit of output variables (namely, CI, ECI, REC, RE, and PN) by leveraging input factors and molecular values, which were subsequently subjected to analysis through the prism of machine learning (ML) algorithms. The aptitude of ML algorithms in the assessment and validation of projected output variables finds resonance in their inherent capacity to incorporate the input parameters underlying the outcomes [24,25,59,60]. Notably, the past years have witnessed the burgeoning utilization of ML models for data validation within *in vitro* regeneration research, manifested across a diverse array of inquiries characterized by distinct objectives and focal points [25,27]. These investigations have embraced a heterogeneous spectrum of models, hyper-parameters, and performance metrics, effectively spanning the gamut of plant tissue culture processes encompassing *in vitro* germination [59,60], somatic embryogenesis [61], *in vitro* sterilization [27], *in vitro* mutagenesis [29], *in vitro* shoot regeneration [22], and the optimization of basal medium [62]. To the best of our knowledge, this study marks a seminal endeavor, pioneering the integration of machine learning (ML) algorithms in conjunction with both polyamines and their corresponding concentrations as input variables. The ascertained outcomes of our investigation underscore the burgeoning potential and steadfast performance of XGBoost in the domain of modeling and prognostication pertinent to the parameters of CI, RE, and PN. Analogously, RF emerged as a promising contender, yielding favorable outcomes in modeling ECI, while SVM exhibited its prowess as a potent algorithm primed for REC. Notably, within the realm of wheat, the XGBoost algorithm emerged as the most adept in modeling three out of the five observed parameters. Drawing parallels to cognate studies, the efficacy of XGBoost in prognosticating chickpea shoot counts exemplifies its capacity to demonstrate superior modeling proficiency [28]. Analogously, RF and XGBoost emerged as apt contenders for forecasting shoot counts and shoot length in *Alternanthera reineckii* [25]. The work by Aasim et al. [59] highlighted the prowess of *Cannabis sativa* L. in engendering an optimal model via the RF algorithm, catering to the prediction and validation of germination and growth indices. In conclusion, the findings collectively advocate for a computational paradigm wherein the synergy of XGBoost, SVM, and RF can manifest as a viable modus operandi for the prediction and optimization of select *in vitro* parameters pertaining to wheat.

## 4. Materials and Methods

### 4.1. In Vitro Conditions

#### 4.1.1. Plant Material, Callus Initiation, Embryogenic Callus Formation, and Plant Regeneration

Seeds of the Kırik wheat (*Triticum aestivum* L.) variety were used as test material. First, the seeds were surface sterilized (in 70% (*v*/*v*) ethanol) for five minutes, rinsed twice with sterile distilled water, incubated in bleach containing a few drops of Tween (5% sodium hypochlorite) for twenty-five minutes, and rinsed twice with sterile distilled water. The seeds were then soaked in sterile water (at 4 °C) for 16–17 h. The shoot axes of mature embryos, as well as their scales, were trimmed both longitudinally and horizontally without detaching them from the seeds. The prepared seeds were cultured on callus induction medium [13]. The callus induction media consisted of Murashige and Skoog [63] base salts, 12 mg L^–1^ dicamba (3,6-dichloro-2-methoxybenzoic acid), 0.5 mg L^–1^ IAA (Indole-3-acetic acid), 20 g L^–1^ sucrose, 2 g L^–1^ phytagel, 1.95 g L^–1^ MES, and different polyamine types (Put: putrescine, Spd: spermidine, and Spm: spermine) and concentrations (0, 0.5, 1, and 1.5 mM). The pH was adjusted with 1 N NaOH to a final value of 5.8 in all media. To sterilize the media solutions, they were autoclaved for 15 min at 121 °C. The solutions contained basic salts and a solidifying agent. Filter sterilization was used for vitamins and plant growth regulators. The percentage of callus induction was determined 21 days after the explants were grown at 25 °C in the dark. The mature callus was then isolated from the seeds. A number of seedlings responding to different types and concentrations of polyamines, callus induction (CI) (%), embryogenic callus induction (ECI) (%), the ratio of responding embryogenic calluses (RECs) (%), and regeneration efficiency (RE) were determined. After four weeks, the ratios of responded embryogenic calluses (RECs) and regeneration efficiency (RE) were determined. Number of callus/number of explants (×100) was used to calculate CI (%). Number of embryogenic callus/number of explant (×100) was used to calculate REC (%) and number of responded embryogenic callus/number of explant (×100) was used to calculate REC%. The number of regenerated plants/number of RECs was used to calculate RE.

The media required for plant regeneration and embryonic calluses were placed in MS medium containing 0.5 mg L^–1^ TDZ (N-Phenyl-N’-1,2,3-thiadiazol-5-ylurea, Thidiazuron), 20 g L^–1^ sucrose, and 7 g L^–1^ agar. They were then cultured at 25 °C for 30 days with a photoperiod of 16 h of light (light intensity 62 mol m^–2^ s^–1^) and 8 h of darkness. After 4 weeks, the number of regenerating plants from each explant was counted. When the regenerating plants reached a height of 10–12 cm, they were transplanted into magenta boxes containing the same regeneration medium and maintained at the same plant regeneration settings.

#### 4.1.2. Statistical Analysis

The present experiments were performed using a factorial randomized complete block design [3 different polyamines (Put, Spd, Spm) × 4 different concentrations of polyamine (0, 0.5, 1 and 1.5 mM)] with four replications and ten explants per replicate. Each Petri dish was considered as experimental unit, and ten mature embryos were cultured on each dish. The Genstat v. 23 statistical software package [64] was used to perform an analysis of variance (ANOVA) using a general linear model procedure. Treatment means and interactions were compared using Duncan’s multiple range test.

### 4.2. Genotypic Assay

#### 4.2.1. Isolation of Genomic DNA, iPBS, and CRED–iPBS PCR Assays and Electrophoresis

According to Zeinalzadehtabrizi et al. [65], genomic DNA was isolated from cultured reactive embryogenic callus using cetyltrimethylammonium bromide buffer [66]. The concentration and purity of genomic DNA were assessed on an agarose gel containing 1.5% (*w*/*v*) using a Nanodrop spectrophotometer (Thermo Fisher Scientific (Waltham, MA, USA)). The PCR reaction for iPBS analysis was carried out using 20 mL PCR mixture containing 50 ng mL^–1^ DNA sample, 1 U Taq DNA polymerase, 10 pmol random primer, 25 mM MgCl_2_, 10 mM dNTP mix, ddH_2_O, and 10 X PCR buffer. The tubes were vortexed and then amplified in a thermocycler (Sensoquest GmbH, Labcycler Gradient, Germany). The PCR protocol included a 5-min initial denaturation at 95 °C, 40 cycles of 1-min denaturation at 95 °C, 1-min annealing at 41.4–49.9 °C, 2-min extension at 72 °C, and a 10-min final primer extension at 72 °C, followed by a drop to 4 °C. CRED–iPBS analysis required digestion of 1 mg of DNA samples from each treatment with 1 L (1 FDU) *Hpa* II (Thermo Scientific) and 1 L (1 FDU) *Msp* I (Thermo Scientific) at 37 °C for 2 h. Nondigested gDNA was replaced by DNA, which was digested by the appropriate endonuclease. Amplification was performed using the primers shown in Table 5. PCR procedures were identical to those used in iPBS analysis [15,16,17,18]. Electrophoresis in 1X SB buffer at 100 V for 120 min with a 100–1000 bp DNA ladder (Sigma Aldrich No: P1473-1VL) was used to determine the approximate molecular weight of iPBSs and CRED–iPBS PCR products separated on a 1.50% agarose gel containing 0.05 mL mL^–1^ ethidium bromide. The gels were fixed in a Universal Hood II (Bio Rad, Hercules, CA, USA) illuminated by ultraviolet light [14].

#### 4.2.2. Genetics Analysis

TotalLab TL120 (Non-linear Dynamics Ltd. R brand, Northumberland, UK) was used to evaluate iPBS and CRED–iPBS banding patterns. The iPBS profiles showed polymorphism when the expected band disappeared and a new band appeared compared to the control. Changes in average polymorphism were determined for each treatment group (different polyamine concentrations) and expressed as a percentage relative to the value obtained from the control (set to 100%) [19]. Genomic template stability (GTS%), a quantitative measure, was calculated for iPBS using the formula (1 − *a*/*n*) × 100, where *a* is the average number of polymorphic bands in each treated template and *n* is the total number of bands in the control [14,19]. The average polymorphism values (%) for each concentration were determined using the formula 100 *a*/*n* [16].

### 4.3. Modeling Using Machine Learning Algorithms

Machine learning (ML) [30], random forest [31], and extreme gradient boosting [32] algorithms were used to train the model and estimate the output variables of tissue culture parameters (callus induction, embryogenic callus induction, embryogenic callus rate, regeneration efficiency, and number of seedlings). In the data set for this study, the inputs that were used included three different experimental polyamines (Put, Spd, and Spm) at different concentrations (0, 0.5, 1 and 1.5 mM). In addition, the results of GTS indices, *Msp* I polymorphism and *Hpa* II polymorphism in wheat as a result of the influence of these experimental groups, were also included in the data set. Estimates on the observed variables (CI, ECI, REC, RE, and PN) were therefore obtained as a result of the influence of both exogenously applied polyamines and changes in gDNA. The wheat dataset was divided into two separate datasets, namely, the training and test sets, with a partition ratio of 70% and 30%, respectively. A leave-one-out cross-validation (LOO-CV) method was used so that the performance of the models could be assessed [67]. A total of four different performance indicators (R^2^, MSE, MAPE, and MAD) were used to evaluate the effectiveness of each model. The coefficient of determination (R^2^) quantifies the degree of correlation between the model and the dependent variables (Equation (1)). Mean squared error (MSE) is a statistical measure used to assess the accuracy of a regression model by quantifying the mean squared difference between predicted and actual values. The mean squared error (MSE) is calculated as the average of the squared deviations between observed and predicted values (Equation (2)). Mean absolute percentage error (MAPE) is a commonly used measure of error in regression analysis that quantifies the extent to which model predictions deviate from the actual values, expressed as a percentage of true values. Mean absolute percentage error (MAPE) calculates the average of the absolute differences between predicted and actual values for each data point, expressed as a percentage of the actual value. Mean absolute percentage error (MAPE) has the property of scale independence, meaning that changes are represented as percentages, making it possible to evaluate different data sets. This feature allows comparison of different data sets (Equation (3)). Mean absolute deviation (MAD) is calculated by obtaining the absolute difference between the actual value and the predicted value for each data point, and then calculating the average of these absolute differences. The MAD metric denotes the arithmetic mean of the absolute values of the residuals. The mean absolute deviation (MAD) metric shows sensitivity to significant outliers due to the use of absolute differences values (Equation (4)).
(1)$$R^{2}=1-\left(\frac{\sum_{i=1}^{n}{\left(y_{i}-y_{ip}\right)}^{2}}{\sum_{i=1}^{n}{\left(y_{i}-\bar{y}\right)}^{2}}\right)$$
(2)$$MSE=\sqrt{\frac{1}{n}\sum_{i=1}^{n}{\left(y_{i}-y_{ip}\right)}^{2}}$$
(3)$$MAPE=\frac{1}{n}\sum_{i=1}^{n}\left|\frac{y_{i}-y_{ip}}{y_{i}}\right|\times 100$$
(4)$$MAD=\frac{1}{n}\sum_{i=1}^{n}\left|y_{i}-y_{ip}\right|$$

Here, *n* is the training/testing sample size in the data set, $y_{i}$ is the measured real value, $y_{ip}$ is the predicted value, and $\bar{y}$ is the measured values mean. The R-4.3.1 software was utilized to compute the ML algorithms and performance indicators (R Core Team).

## 5. Conclusions

The success of improving plant species by biotechnological methods depends on the response of the tissue culture, specifically in the regeneration of plants. Research on the effects of polyamines on plant growth and development is widespread. To our knowledge, this is the first study that focused on elucidating the effects of polyamines on DNA methylation *in vitro* using the ML approach. The CRED–iPBS method was also used to determine how different polyamines and doses affect DNA methylation patterns. Changes in DNA methylation at high concentrations of polyamine were also detected. Re-testing showed that the highest RE was obtained with 1.5 mM Put. Additionally, the result clearly showed that 1.5 mM Spd was associated with DNA hypermethylation, while hypomethylation occurred at 0.5 mM Spm. In addition, four different performance indicators (R^2^, MSE, MAPE, and MAD) were used to evaluate the performance of each model. According to the results, a computational technique using a combination of XGBoost, SVM, and RF may be a promising way to predict and improve selected wheat *in vitro* parameters.

## Figures and Tables

**Figure 1 plants-12-03261-f001:**
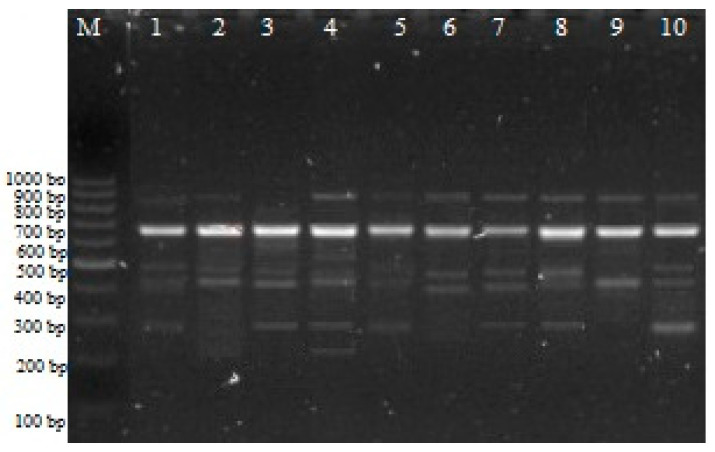
iPBS profiles for various PA experimental groups with 2384 primers. M: 100–1000 bp DNA ladder; 1: control; 2: 0.5 mM Put; 3: 1 mM Put; 4: 1.5 mM Put; 5: 0.5 mM Spd; 6: 1 mM Spd;7: 0.5 mM Spd; 8: 0.5 mM Spm; 9: 1 µM Spm; 10: 1.5 mM Spm.

**Figure 2 plants-12-03261-f002:**
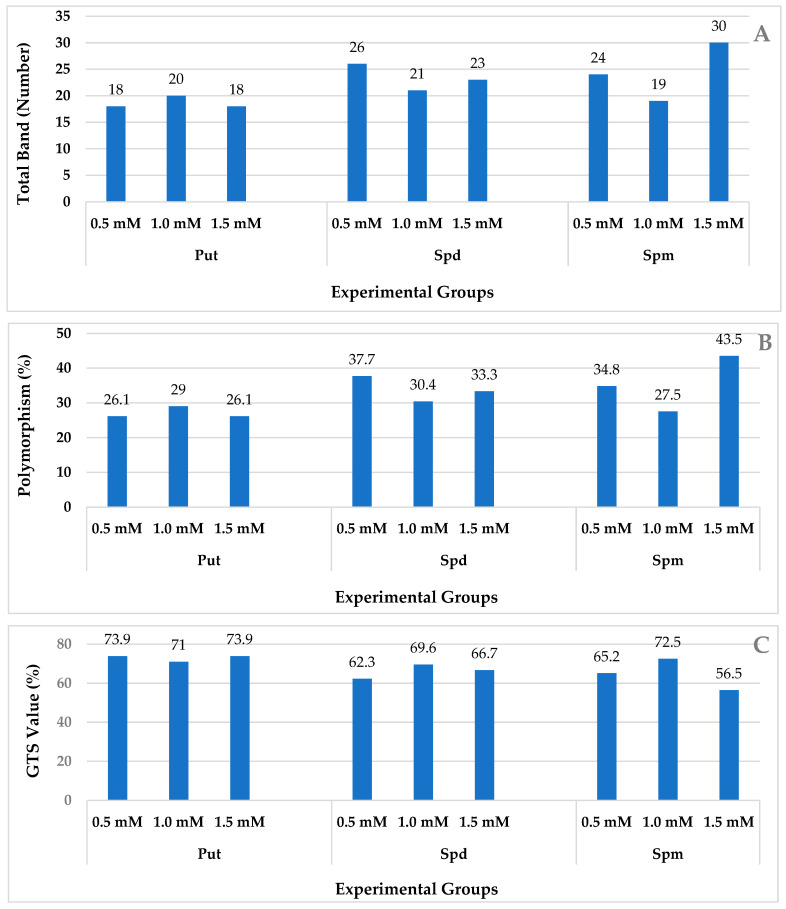
DNA methylation changes in the wheat exposed to PAs: (**A**) total band; (**B**) polymorphism; (**C**) GTS value as estimated using different MSH experimental groups.

**Figure 3 plants-12-03261-f003:**
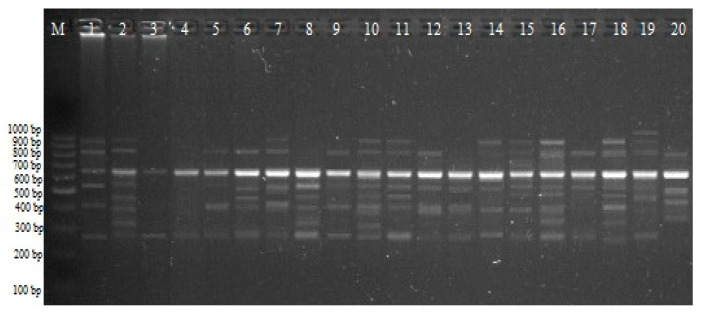
CRED–iPBS profiles for various PA experimental groups with iPBS 2384 primers; M, 100–1000 bp DNA ladder; 1: control *Hpa* II; 2: control *Msp* I; 3: 0.5 mM Put *Hpa* II; 4: 0.5 mM Put *Msp* I; 5: 1 mM Put *Hpa* II; 6: 1 mM Put *Msp* I; 7: 1.5 mM Put *Hpa* II; 8: 1.5 mM Put *Msp* I; 9: 0.5 mM Spd *Hpa* II; 10: 0.5 mM Spd *Msp* I; 11: 1 mM Spd *Hpa* II; 12: 1 mM Spd *Msp* I; 13: 1.5 mM Spd *Hpa* II; 14: 1.5 mM Spd *Msp* I; 15: 0.5 mM Spm *Hpa* II; 16: 0.5 mM Spm *Msp* I; 17: 1 mM Spm *Hpa* II; 18: 1 mM Spm *Msp* I; 19: 1.5 mM Spm *Hpa* II; 20: 1.5 mM Spm *Msp* I.

**Figure 4 plants-12-03261-f004:**
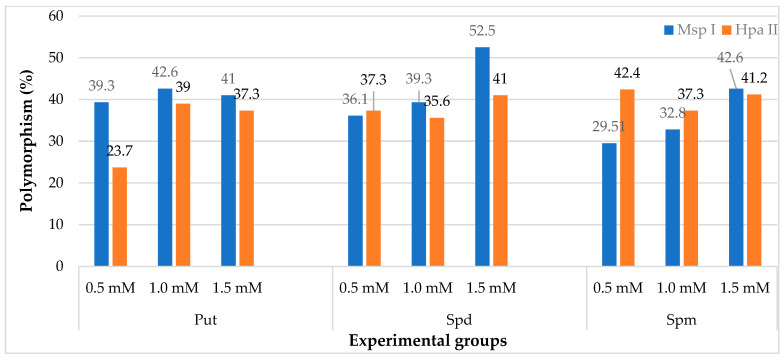
The effect of PAs on polymorphism percentages in different experimental groups of wheat in the seedling growth stage.

**Table 1 plants-12-03261-t001:** Callus induction (CI) (%), embryogenic callus induction (ECI) (%), responsive embryogenic callus (REC) (%), regeneration efficiency (RE), and plantlet number (PN) (number) responses to different polyamine types and their concentrations.

Polyamine Types	Concentration (mM)	CI% ^1^	ECI% ^2^	REC% ^3^	RE ^4^	PN ^5^
Putrescine (Put)	0	92.50 ^abcd 6^	77.50 ^bcd^	77.50 ^ab^	1.15 ^abc^	7.75 ^b^
	0.5	95.00 ^ab^	79.30 ^bcd^	89.30 ^a^	1.14 ^abc^	8.50 ^b^
	1	97.50 ^a^	91.80 ^ab^	65.00 ^bc^	1.17 ^abc^	12.25 ^a^
	1.5	90.00 ^bcde^	85.00 ^abcd^	50.60 ^cde^	1.29 ^ab^	3.00 ^de^
	Means	93.75 ^A^	83.43 ^A^	70.62 ^A^	1.19 ^A^	7.87 ^A^
Spermidine (Spd)	0	92.50 ^abcd^	77.50 ^bcd^	77.50 ^ab^	1.15 ^abc^	7.75 ^b^
	0.5	93.75 ^abc^	77.50 ^bcd^	82.50 ^a^	0.95 ^c^	8.75 ^b^
	1	92.50 ^abcd^	87.50 ^abc^	92.50 ^a^	1.09 ^bc^	12.00 ^a^
	1.5	92.50 ^abcd^	100.00 ^a^	57.50 ^cd^	1.07 ^bc^	2.50 ^e^
	Means	92.812 ^A^	85.62 ^A^	77.50 ^A^	1.07 ^AB^	7.75 ^A^
Spermine (Spm)	0	92.50 ^abcd^	77.50 ^bcd^	77.50 ^ab^	1.15 ^abc^	7.75 ^b^
	0.5	87.50 ^cde^	67.50 ^d^	47.50 ^de^	1.01 ^bc^	5.25 ^c^
	1	83.75 ^e^	70.00 ^cd^	45.00 ^de^	1.40 ^a^	4.50 ^cd^
	1.5	86.25 ^de^	67.50 ^d^	35.00 ^e^	0.37 ^d^	2.00 ^e^
	Means	87.50 ^B^	70.625 ^B^	51.25 ^B^	0.98 ^B^	4.87 ^B^
Mean square of polyamine (P)		181.77 **	1050.52 **	2964.58 **	0.17 **	46.08 **
The mean square of concentration (C)		19.96 ^ns^	241.84 ^ns^	2076.04 **	0.22 **	110.5 **
The mean square of P × C		38.71 ^ns^	235.59 ^ns^	619.79 **	0.29 **	15.91 **

** significant at *p* ≤ 0.01, ^ns^ non-significant at *p* ≥ 0.05. ^2^ The means of columns, rows, and factors all beginning with the same letter do not vary statistically (*p* > 0.05). ^1^ (Callus number/explant number) × 100. ^2^ (Embryogenic callus number/explant number) × 100. ^3^ (Responded embryogenic callus number/explant number) × 100. ^4^ (Number of Regenerated Plants/Number of RECs. ^5^ (number of regenerating plants from each explant). ^6^ Upper and lower-case letters denote items of importance.

**Table 2 plants-12-03261-t002:** Molecular sizes (bp) of bands present/absent in profiles of wheat genotypes with different polyamines and their concentrations.

iPBS Primer	± ^1^	Control ^2^	Experimental Groups								
			Putresince (Put)			Spermidine (Spd)			Spermine (Spm)		
			0.5 mM	1.0 mM	1.5 mM	0.5 mM	1.0 mM	1.5 mM	0.5 mM	1.0 mM	1.5 mM
2075	+	6	-	590	689; 590	590	590	-	571	-	461
	-		-	-	-	-	-	-	-	-	-
2077	+	6	-	1120; 918; 659; 467; 372	379	1600; 1100; 900; 475	1140; 843; 379	1600; 1060; 957; 475; 379	491; 379	-	1600; 1080
	-		813; 700	-	-	-	-	-	-	-	311
2087	+	4	-	1600; 452; 416	-	1100; 480	-	425	-	-	1340
	-		-	-	-	727; 648	831; 500	-	831; 727; 500	831; 500	727; 648; 500
2278	+	9	-	1200; 950	1250	1357	1357	1320; 1200; 950	1320	1320; 1200	1320; 1200
	-		752; 520	-	520	520	752	-	520	-	752; 400; 310
2375	+	10	-	1371; 970; 852; 536	1371	1357	1357	1375; 1285; 1200; 1157; 985; 575	1371	1375; 1285; 1157	1375; 1285
	-		779; 607	-	607	607	779	-	607	-	779; 718; 486; 308
2377	+	9	-	-	-	-	-	-	-	-	1214
	-		1057; 970; 710; 582; 431	431	970; 582; 487; 431	582; 487; 431	900; 710; 637; 582; 520; 487; 431	487; 431	487; 431	710; 487; 431	487; 431
2380	+	5	-	-	548	1060; 837	-	1060; 920	1040; 858	858; 560	-
	-		-	-	-	-	-	-	-	-	-
2381	+	9	-	-	-	-	-	-	-	-	-
	-		1300; 1200; 833; 420; 376	472; 376	1300; 1200; 972; 833	1300; 1200; 972; 472; 420; 324	972; 376	472; 376	1300; 1200; 972; 833; 651; 472; 420; 376	1200; 833; 472; 376	1300; 1200; 833; 472; 376
2382	+	6	936	917	-	1000	-	1040; 413	-	-	-
	-		-	-	-	521	521	-	589; 500	521	870; 589; 521
2384	+	5	-	-	525; 239	-	-	-	-	-	-
	-		282	800	-	-	282	-	-	392; 282	-

^1^, ^2^ appearance of a new band (+), disappearance of a normal band (-), and without PAs, respectively.

**Table 3 plants-12-03261-t003:** Results of CRED–iPBS analysis; molecular size of bands and polymorphism percentage.

iPBS Primer	M/H ^1^		± ^2^	Control ^3^	Experimental Groups								
					Putresince (Put)			Spermidine (Spd)			Spermine (Spm)		
					0.5 mM	1.0 mM	1.5 mM	0.5 mM	1.0 mM	1.5 mM	0.5 mM	1.0 mM	1.5 mM
2075	M		+	7	-	-	-	-	-	-	315	-	-
			-		393	-	-	-	-	393	-	393; 271	271
	H		+	7	-	-	376	-	-	-	626	-	-
			-		400	-	-	-	400	400	-	-	266; 180
2077	M		+	6	465	1416; 1216; 1033; 918; 465	1416; 1216	1433; 1050; 918	1416; 1016; 918	1416; 1016; 900; 613; 491; 377	1416; 1016; 880	1413; 1016	1433; 1033
			-		-	-	-	575; 415; 356; 312	415; 312		415; 312	415; 312	312
	H		+	6	818	1450; 1033; 936	1433; 1033; 918	1416; 1183; 1050	1433; 1050	1433; 1033; 984; 900; 800; 739	1433; 1300; 1033;	1416; 1000; 918	1433; 1266; 1016
			-		-	-	600	356; 318	318	-	-	318	415; 318
2087	M		+	7	522; 400; 350	-	-	-	-	-	-	-	-
			-		1300; 1075	1300; 1075; 815	1075; 922; 815; 700; 600; 489	1075; 922; 815; 700; 489	1300; 1075; 922; 700; 600;	1075; 922; 815; 700; 600;	1300; 1075; 922; 700; 600	1075; 922; 815; 600;	1300; 1075; 922; 815; 700; 600
	H		+	8	1100; 427	1250	-	-	-	-	-	-	-
			-		-	700; 567; 418	1125; 629; 418	922; 815; 700; 567; 500	1125; 922; 815; 629; 567; 418	815; 567; 500; 418	1125; 922; 815;700; 629; 418	1125; 922; 815; 629; 567; 418	1125; 922; 815; 629; 567; 418
2278		M	+	8	500	-	-	-	-	-	-	-	-
			-		1100; 900	1100; 900	900; 750; 650; 450	900; 815; 750; 450	1100; 900; 750; 450	900; 815; 750; 450	1100; 900; 750	900; 815;	1100; 900; 815; 750
		H	+	5	1100; 427	1250	-	-	-	-	-	-	-
			-		-	760; 465	1125; 465	1125; 760	760; 465	1125; 760; 465	-	1125;465	1125; 760
2375		M	+	5	1040	1100; 851; 811	1100; 866	-	837	851	-	938; 811	824; 326
			-		-	-	-	405	-	-	-	-	-
		H	+	4	-	1280; 1060; 589	1040; 589; 468	1020; 589	1020; 454	1100; 1020; 589	1080; 1020; 651; 589	1020	1020; 680; 365
			-		866;811; 709	-	-	-	709	-	-	-	-
2377		M	+	6	640; 583; 492; 338; 264	964; 739; 682; 632	913	-	926	1075; 939; 762; 648; 616; 492	1100	1087	1112
			-		-	-	-	-	-	-	-	548	400
		H	+	6	-	1260; 1087; 762	1262	1262	-	1275; 1087	-	1287	1275
			-		-	-	431; 394	431; 394	431	-	-	431; 394	729; 500
2380	M		+	5	-	607	732; 607; 568; 450	875; 607; 527	607	944; 880; 613; 580	-	961	912
			-		-	-	-	-	-	-	-	400	-
	H		+	5	-	673	600; 500	607; 500	-	981	944; 527	1000; 555	591
			-		-	846	-	-	400	-	-	-	-
2381	M		+	5	-	1325; 1025; 750; 334	-	-	-	-	-		325; 269
			-		-	-	833; 466	537	921; 633; 537	466	921; 633; 466; 409	537; 466; 409	921; 633
	H		+	4	-	1325; 1100; 679	900; 679	-	-	-	-	-	-
			-		-	-	-	611	427	-	819; 427; 353	819; 353	819; 427; 353
2382	M		+	4	753; 529	1016; 848	586; 369	408	-	545	529	-	-
			-		-	-	-	-	492	-	-	-	-
	H		+	8	-	-	-	-	-	-	-	-	-
			-		940	940;	753; 645	645	940; 537	940; 753	940; 753; 645; 537	-	940; 537
2384	M		+	8	-	-	-	-	-	-	-	-	-
			-		958; 830; 540; 461; 391; 346	958; 346	958; 830	-	958; 461; 346	830; 461; 346	-	-	958; 346; 281
	H		+	6	-	-	-	-	461	-	700; 480	-	1020; 452
			-		958; 830; 551; 408	958	-	958	-	958; 830	-	958; 408	-

^1, 2^ and ^3^: M—*Msp* I and H—*Hpa* II; (+) appearance of a new band, (-) disappearance of a normal band; and without hormone, respectively.

**Table 4 plants-12-03261-t004:** The goodness of fit criteria for machine learning algorithms.

Traits ^1^	ML Criteria ^2^	SVM		RF		XGBoost	
		Test	Train	Test	Train	Test	Train
CI (%)	R^2^	0.254	0.407	0.379	0.486	0.383	0.529
	MSE	4.364	4.121	3.981	3.838	3.889	3.673
	MAPE	3.816	3.401	3.662	3.204	3.289	3.057
	MAD	3.540	3.958	3.386	3.136	3.243	2.841
ECI (%)	R^2^	0.351	0.312	0.415	0.48	0.231	0.501
	MSE	9.744	11.160	9.245	9.005	9.608	9.800
	MAPE	10.535	11.202	9.435	10.217	9.473	10.243
	MAD	7.624	7.528	7.063	6.087	7.064	6.977
REC (%)	R^2^	0.841	0.776	0.781	0.792	0.699	0.812
	MSE	9.699	9.336	10.367	9.386	10.459	9.918
	MAPE	14.405	12.603	15.691	12.653	16.585	12.690
	MAD	7.197	6.950	7.783	7.184	8.282	7.490
RE	R^2^	0.482	0.356	0.646	0.548	0.738	0.613
	MSE	0.271	0.198	0.224	0.166	0.193	0.154
	MAPE	42.415	15.433	34.323	14.739	23.748	11.776
	MAD	0.219	0.125	0.185	0.120	0.165	0.110
PN	R^2^	0.782	0.833	0.775	0.880	0.853	0.905
	MSE	1.761	1.363	1.791	1.155	1.445	1.029
	MAPE	19.548	24.429	17.156	22.011	11.653	18.343
	MAD	1.137	0.899	1.192	0.815	0.926	0.770

^1^ CI (%); Callus induction, ECI (%); embryogenic callus induction, REC (%); responding embryogenic callus and regeneration efficiency (RE) and plantlet number (PN) responding to different polyamine types and concentrations. ^2^ R^2^: Coefficient of determination, mean squared error (MSE), mean absolute percentage error (MAPE), mean absolute deviation (MAD).

**Table 5 plants-12-03261-t005:** Ten iPBS primers’ sequences and annealing (Ta) temperatures are provided.

Primer Name	Sequence (5′–3′)	Tm (°C) ^1^	CG (%) ^2^
iPBS-2075	CTCATGATGCCA	42.1	50.0
iPBS-2077	CTCACGATGCCA	46.1	58.3
iPBS-2087	GCAATGGAACCA	43.5	50.0
iPBS-2278	GCTCATGATACCA	42.3	46.2
iPBS-2375	TCGCATCAACCA	45.1	50.0
iPBS-2377	ACGAAGGGACCA	47.2	58.3
iPBS-2380	CAACCTGATCCA	41.4	50.0
iPBS-2381	GTCCATCTTCCA	49.9	50.0
iPBS-2382	TGTTGGCTTCCA	44.9	50.0
iPBS-2384	GTAATGGGTCCA	40.9	50.0

^1^, ^2^; Tm: primer temperatures; CG: percentage of cytosine (C) and guanine (G) in the primary sequence, respectively.

## Data Availability

All data supporting the conclusions of this article are included in this article.

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
