# Peer review of "Investigation of the Influence of Polyamines on Mature Embryo Culture and DNA Methylation of Wheat (Triticum aestivum L.) Using the Machine Learning Algorithm Method"

_plants, 2023, doi:10.3390/plants12183261_

Round 1
Reviewer 1 Report
The article needs to be reorganized and restructured. At present it lacks a coherent structure. The abstract and conclusion sections are unnecessarily lengthy and lack significance. Even the title is too verbose.
The results are not described well. The results are compiled based on the analysis/methodology, instead of significance and outcomes.
English needs attention.
Author Response
Responses to Comments of Reviewer 1
General Response:
First of all, we thank the potential reviewer for her/his valuable time and also raised helpful comments and suggestions. In this step of revision, we have tried to respond to all comments and addressed all questions. We hope the revised version of manuscript gets positive feedback from you and will be acceptable for publication in the Plants journal. All revised parts have been highlighted in blue.
Sincerely,
Dr. Aras Turkoglu
Comments
Comment #1:
The article needs to be reorganized and restructured. At present it lacks a coherent structure. The abstract and conclusion sections are unnecessarily lengthy and lack significance. Even the title is too verbose.
Response to Comment 1#
Dear reviewer we added new link sentences among issues also restructured the main text again, especially we summarized the title, abstract and conclusion sections.
Comment #2:
The results are not described well. The results are compiled based on the analysis/methodology, instead of significance and outcomes.
Response to Comment 2#
Dear reviewer we added necessary sentences this section as well as we completely explain reason of the results in the discussion.

Reviewer 2 Report
This manuscript described the use of a machine learning algorithm to understand callus formation and regeneration with the use of different polyamines. They used XGBoost method in wheat to identify the value of callus induction, regeneration efficiency and number of responsive seedlings. Overall, the study is interesting however I observe some lack of connectivity to readers mainly due to the way data is presented, and the lack of explanation of the importance and relevance of the presented research.
Starting with the title- it is very long and includes too much information- I counted at least five topics (modelling, regeneration, polyamines, ML, DNA damage, methylation and LTR polymorphism). It would be better if authors could make precise and shorter title reflecting the core message/s.
On the contrary, the abstract, which needs to be precise and informative, explainable for important information of the manuscript, is not focussing on the core message. It rather includes a lengthy introduction along with many of the results. Rewriting of the abstract is highly recommended, along with some good English corrections.
Vis-a-vis English correction, in some places , sentences are very long and confusing, and language requires some editing. Here are a few examples (see more below under minor comments):
Ø line 66-69, 99-105, 128-130. Line 88-89, end with ‘and etc’. Is very confusing and inappropriate.
Ø Line 141, MS medium should be elaborative in first use. Later authors can use abbreviations; this will apply to the whole manuscript.
Ø And there is more—it is recommended that English editing will be conducted on the whole manuscript, including the abstract, results and discussion.
The results part goes directly to the description of the main finding without explaining why and how these experiments were performed. This is a general flaw of the manuscript, and I recommend giving some rationale to each part of the results they present before explaining what they got.
In that regard, I could not quite understand why the authors tested and determined the genetic and epigenetic effects of different polyamine types and concentrations in wheat plants using iPBS and CRED-iPBS methods. I did find this later in the discussion (lines 84-86), “Point mutations, insertions, and deletions at loci produced by the markers employed are possible causes of the regeneration”. If so, wouldn’t it make sense to test this somehow by perhaps associating certain polymorphism with the quantitative data they have (of regeneration). Otherwise, this is very descriptive data presented as very large tables. Perhaps I am missing something, but once they explain better why they did that, I suggest that this data will not be presented as it is now. If they decide to present these, then tables 3 and 4 are non-friendly to the reader and should be replaced with figures.
Table 1 describes the percentages of regeneration in the different treatments, yet it would help to know the number of embryos used in these experiments. In addition, what exactly does RE (regeneration efficiency) stand for? This is since in the table, it says “numbers” in brackets—what do these stand for?
I would recommend starting the discussion at line 11 of the third page 14, and consider if including the part before that is required. This is because it is a paragraph composed of several ideas that do not bind together well enough to begin the discussion.
Minor comments
Ø Any abbreviation that appears for the first time should be spelt out. For example, iPBS (line 103), CRED (line 110)
More important is in the case of MSE, MAPE and MAD (line 189), which are critical for getting a better understanding of the estimators of machine learning
Ø Line 10—remove the “are”
Ø Table-1: Mean square of poliamine (P) to “Mean square of polyamine (P)”
Ø Line 45: should have 85.3% (not only 85.3)
To conclude, the potential of these studies is substantial as it may assist non-expert in the field of ML to utilize these tools for improving regeneration and genetic editing efforts in a major crop such as wheat and other reluctant crop plants. Nevertheless, it would have helped have the authors present the data less descriptively, and instead of large tables that are hard to follow, could explain results as figures.
Please see within comments.
Author Response
Responses to Comments of Reviewer 2
General Response:
First of all, we thank the potential reviewer for her/his valuable time and also raised helpful comments and suggestions. In this step of revision, we have tried to respond to all comments and addressed all questions. We hope the revised version of manuscript gets positive feedback from you and will be acceptable for publication in the Plants journal. All revised parts have been highlighted in blue.
Sincerely,
Dr. Aras Turkoglu
Comments
Comment #1
This manuscript described the use of a machine learning algorithm to understand callus formation and regeneration with the use of different polyamines. They used XGBoost method in wheat to identify the value of callus induction, regeneration efficiency and number of responsive seedlings. Overall, the study is interesting however I observe some lack of connectivity to readers mainly due to the way data is presented, and the lack of explanation of the importance and relevance of the presented research.
Response to Comment 1#
Dear reviewer we added new link sentences among issues also restructured the main text again.
Comment #2
Starting with the title- it is very long and includes too much information- I counted at least five topics (modelling, regeneration, polyamines, ML, DNA damage, methylation and LTR polymorphism). It would be better if authors could make precise and shorter title reflecting the core message/s.
Response to Comment 2#
We rewritten new the title; new title is short as well as and represents the work.
Comment #3
On the contrary, the abstract, which needs to be precise and informative, explainable for important information of the manuscript, is not focusing on the core message. It rather includes a lengthy introduction along with many of the results. Rewriting of the abstract is highly recommended, along with some good English corrections.
Response to Comment 3#
We have rewritten and shortened the abstract and tried to follow the suggested recommendations. To polish the language and eliminate errors, we have revised the text by a native.
Comment #4
Vis-a-vis English correction, in some places, sentences are very long and confusing, and language requires some editing. Here are a few examples (see more below under minor comments)
Response to Comment 4#
To polish the language and eliminate errors, we revised the text by a native and illustrated it with track change.
Comment #5
Ø line 66-69, 99-105, 128-130. Line 88-89, end with ‘and etc’. Is very confusing and inappropriate.
Response to Comment 5#
We have rewritten this part.
Comment #6
Ø Line 141, MS medium should be elaborative in first use. Later authors can use abbreviations; this will apply to the whole manuscript.
Response to Comment 6#
We have rewritten this part.
Comment #7
Ø And there is more—it is recommended that English editing will be conducted on the whole manuscript, including the abstract, results and discussion.
Response to Comment 7#
To polish the language and eliminate errors, we revised the text by a native and illustrated it with track change.
Comment #8
The results part goes directly to the description of the main finding without explaining why and how these experiments were performed. This is a general flaw of the manuscript, and I recommend giving some rationale to each part of the results they present before explaining what they got.
Response to Comment 8#
We thank the reviewer to bold these issues. In the revised text, we have addressed to these topics.
Comment #9
In that regard, I could not quite understand why the authors tested and determined the genetic and epigenetic effects of different polyamine types and concentrations in wheat plants using iPBS and CRED-iPBS methods. I did find this later in the discussion (lines 84-86), “Point mutations, insertions, and deletions at loci produced by the markers employed are possible causes of the regeneration”. If so, wouldn’t it make sense to test this somehow by perhaps associating certain polymorphism with the quantitative data they have (of regeneration). Otherwise, this is very descriptive data presented as very large tables. Perhaps I am missing something, but once they explain better why they did that, I suggest that this data will not be presented as it is now. If they decide to present these, then tables 3 and 4 are non-friendly to the reader and should be replaced with figures.
Response to Comment 9#
Dear reviewer we added new link sentences among issues also restructured the main text again, especially we have added the related analysis. However, we used them as Figure 1, 2, 3 and 4 based on the MDPI style.
Comment #10
Table 1 describes the percentages of regeneration in the different treatments, yet it would help to know the number of embryos used in these experiments. In addition, what exactly does RE (regeneration efficiency) stand for? This is since in the table, it says “numbers” in brackets—what do these stands for?
Response to Comment 10#
The sentences of ~Each petri dish was considered as an experimental unit and 10 mature embryos were cultured in each petri dish~ was added the material and methods section, as well as we added the abbreviation under the table also, regeneration efficiency information whole the text.
Comment #11
I would recommend starting the discussion at line 11 of the third page 14, and consider if including the part before that is required. This is because it is a paragraph composed of several ideas that do not bind together well enough to begin the discussion.
Response to Comment 11#
We thank the reviewer to bold these issues. In the revised text, we have addressed to these topics.
Comment #12
Ø Any abbreviation that appears for the first time should be spelt out. For example, iPBS (line 103), CRED (line 110)
Response to Comment 12#
All abbreviation checked the main text.
Comment #13
More important is in the case of MSE, MAPE and MAD (line 189), which are critical for getting a better understanding of the estimators of machine learning
Response to Comment 13#
All abbreviation checked the main text.
Comment #14
Ø Line 10—remove the “are”
Response to Comment 14#
To polish the language and eliminate errors, we revised the text by a native and illustrated it with track change.
Comment #15
Ø Table-1: Mean square of poliamine (P) to “Mean square of polyamine (P)”
Response to Comment 15#
To polish the language and eliminate errors, we revised the text by a native and illustrated it with track change.
Comment #16
Ø Line 45: should have 85.3% (not only 85.3)
Response to Comment 16#
To polish the language and eliminate errors, we revised the text by a native and illustrated it with track change.
Comment #17
To conclude, the potential of these studies is substantial as it may assist non-expert in the field of ML to utilize these tools for improving regeneration and genetic editing efforts in a major crop such as wheat and other reluctant crop plants. Nevertheless, it would have helped have the authors present the data less descriptively, and instead of large tables that are hard to follow, could explain results as figures.
Response to Comment 17#
We thank the reviewer to bold these issues. In the revised text, we have added the Figure 1,2,3 and 4 in main text for better presentation data and results.

Round 2
Reviewer 1 Report
Authors have revised well.
NA